# Rectal Polyposis in Mucosal Prolapse Syndrome

**DOI:** 10.3390/diagnostics12040966

**Published:** 2022-04-12

**Authors:** Yin Ping Wong, Connie Kabincong, Mohd Faisal Jabar, Geok Chin Tan

**Affiliations:** 1Department of Pathology, Faculty of Medicine, Universiti Kebangsaan Malaysia, Kuala Lumpur 56000, Malaysia; ypwong@ppukm.ukm.edu.my (Y.P.W.); p100756@siswa.ukm.edu.my (C.K.); 2Prince Court Medical Centre, Jalan Kia Peng, Kuala Lumpur 50450, Malaysia; faisal@upm.edu.my

**Keywords:** rectal polyp, mucosal prolapse, histological mimics

## Abstract

Mucosal prolapse syndrome is also known as solitary rectal ulcer syndrome. It may either presents as an ulcer or polyp, which could mimic other pathological lesions such as juvenile polyp, hyperplastic polyp, adenomatous polyp, polyp related inflammatory bowel disease and adenocarcinoma. It can pose as a diagnostic challenge to both the surgeons and pathologists due to the overlapping gross and histological features. The characteristic histological features of mucosal prolapse syndrome are fibromuscular obliteration of lamina propria and splayed hypertrophic muscularis mucosae. It can occur in a wide range of ages, including children and teenagers. Rectal bleeding is one of the common presenting symptoms. Here, we described two cases of mucosal prolapse syndrome presented as rectal polyposis and provide a discussion on its histological differential diagnosis.

## 1. Introduction

Mucosal prolapse syndrome (MPS) is a condition with confusing terms and histological features. It is historically called as solitary rectal ulcer syndrome, even though it is not always solitary, not always in the rectum and not always an ulcer. The incidence of MPS is approximately 1 in 100,000 with young adult predominant. The proposed pathogenesis of SRUS is due to excessive straining which results in high intrarectal pressure and ischaemia. The anterior rectal mucosa is forced into the closed anal canal due to contracting/non-relaxation of puborectalis muscle, leading to congestion and oedema, forming ulcer or polyp. It may present clinically with bleeding, mucus discharge, abdominal pain, tenesmus, feelings of incomplete defaecation and constipation [1].

The gross appearance of either an ulcer or polyp may be mistaken as malignancy to the surgeon. It can be diagnostically challenging to pathologists due to its varied histological appearances. It could mimic other pathological lesions such as juvenile polyp, hyperplastic polyp, tubular adenoma, serrated adenoma, adenocarcinoma, pseudomembranous colitis and inflammatory bowel disease [2]. Polyp is seen in about a quarter of cases in MPS [3]. Here, we described two cases of mucosal prolapse syndrome with rectal polyposis and described the histologically differential diagnosis of MPS with polyp.

## 2. Case Series

### 2.1. CASE 1

#### 2.1.1. Clinical Presentation

An 18-year-old man presented to the emergency unit with one-week history of daily and persistent rectal bleeding. He also noticed a prolapse mass with berries-like surface from the anus, associated with pain. He did not have constipation. He was mildly anaemic with haemoglobin level of 12.8 g/dL. Other laboratory investigations showed total protein level of 69 g/L (normal range 64–83 g/L), albumin level was 43 g/L (normal range 35–52 g/L), globulin level was 26 g/L (normal range 18–42 g/L), and the albumin to globulin ratio was 1.7 (normal range 0.8–2.0). The white cell count was 5.8 × 10^3^/µL (normal range 4.0–11.0 × 10^3^/µL). There was no history of bleeding tendency and his platelet count was 274 × 10^3^/µL. Examination of the rectum showed multiple polyps on the mucosal surface which was confined to the rectum and extend to the distal surgical margin. Altemeier proctosigmoidectomy was performed.

#### 2.1.2. Pathological Features

Gross examination revealed approximately 42 polyps on the mucosal surface (Figure 1A). The largest polyp measured 1.4 cm × 1.0 cm × 0.8 cm. There was no lymph node enlargement in the perirectal tissue. Microscopically, the polyps consisted of granulation tissue with ulcerated surface. There were focal mucosal crypts dilatation, hypertrophic and splayed muscularis mucosa and ectatic submucosal blood vessels (Figure 1B). These features are consistent with mucosal prolapse syndrome. At one-week post-operative follow up, he was well with neither rectal bleeding nor constipation.

### 2.2. CASE 2

#### 2.2.1. Clinical Presentation

A 62-year-old lady with systemic lupus erythematosus presented to our hospital with intermittent watery rectal discharged for the past few years, which became worsened one-month ago. It was associated with constipation and occasional rectal bleeding. Ten years ago, there was a history of total abdominal hysterectomy with bilateral salpingo-oophorectomy. Proctoscopy revealed a pedunculated rectal polyp and grade 4 external haemorrhoid. Polypectomy was performed. Laboratory investigations showed haemoglobin level of 11.3 g/dL (normal range 12.0–15.0 g/dL), white cell count was 6.1 × 10^3^/µL (normal range 4–10 × 10^3^/µL), platelet count level was 243 × 10^3^/µL (normal range 150–410 × 10^3^/µL), c-reactive protein level was 0.14 mg/dL (normal range < 0.5 mg/dL), total protein level was 75 g/L (normal range 64–83 g/L) and albumin level was 42 g/L (normal range 34–48 g/L).

#### 2.2.2. Pathological Features

The rectal polyp measured 2.0 cm × 1.5 cm × 1.3 cm. Microscopically, the polyp consisted of irregularly spaced tubular crypts with focal dilatation, separated by hypertrophied muscularis mucosae, and granulation tissue cap. It was focally covered by squamous epithelium. Some of the glands demonstrate serrations (Figure 2). These features are consistent with mucosal prolapse syndrome. Weigert Van Gieson stain demonstrates elastic fibers in the lamina propria (Figure 3). At 3-week post-polypectomy follow up, she was well with no further watery rectal discharge or bleeding. She was scheduled for hemorrhoidectomy.

## 3. Methods

Weigert Van Gieson (Bio-Optica, 04-051802, Milano, Italy) was used for the staining of elastic fibers. Tissue blocks were sectioned at 4 µm thickness and the tissue slides were incubated with 10 drops of periodic acid solution for 5 min. This was followed by rinsing in distilled water. After that transferred Weigert’s solution into a Coplin Jar, immersed the slides in the jar, covered it, and incubated for overnight. Then, rinsed the slides in distilled water. Subsequently, incubate the slides with 10 drops of acid differentiation buffer for 10 min. This was followed again with rinsing with distilled water. After that, put on the slides 5 drops of Weigert’s iron hematoxylin (solution A) and add 5 drops of Weigert’s iron hematoxylin (solution B) and incubated for 10 min. Then, blue it in running tap water for 10 min. Lastly, put on the slides 10 drops of Van Gieson’s Picrofuchsine and incubated for 7 min. The slides were then rinsed for 2–3 s in distilled water, dehydrate and cover-slipped with mounting medium. TP-XCAM4K8MPA (Matrix Optics, Petaling Jaya, Malaysia) camera mounted on an Olympus microscope BX41 (Olympus Corporation, Tokyo, Japan) was used to obtain the figures of H&E and EVG staining in this study.

## 4. Discussion

MPS was first described by Cruveilhier in 1892 as four unusual cases of rectal ulcers [4]. It is also known as solitary rectal ulcer syndrome (SRUS) and is commonly presented at 3rd to 4th decade in adult. However, it can occur at any age. MPS is a spectrum of diseases that include SRUS, inflammatory cloacogenic polyp, proctitis cystica profunda, inflammatory myoglandular polyp and inflammatory cap polyposis [5]. It may have varied appearance, such as a solitary ulcer, multiple ulcers, patchy granular erythematous mucosa or polypoidal lesion [6]. SRUS is a misnomer as the ulcer can be multiple and can even involve the sigmoid colon. The polypoid appearance may be mistaken as colorectal polyposis syndrome; hence, this entity must be recognised to prevent inappropriate treatment [7]. The characteristic histological features of MPS are fibromuscular obliteration of lamina propria and splayed hypertrophic muscularis mucosae. Therefore, it is essential that the biopsy sample should have sufficient depth. As seen in our cases, the polyps can entirely consist of granulation tissue or a mixture of glands and granulation tissue. The granulation tissue polyp may protrude beyond the mucosa. As a consequence, rectal bleeding is one of the symptoms and could lead to anaemia depending on the severity of blood loss [8,9].

The second case showed the typical histological features of a mucosal prolapse syndrome with cystically dilated glands, some of them demonstrate intraluminal serrations, villiform surface, exudate on the mucosal surface simulating pseudomembrane, and thickened and disorganised muscularis mucosa (see Figure 2). Addition histological features of MPS are glands within the muscularis mucosa and submucosa, and thickened blood vessels with fibrin in the vessel wall [10].

One of our cases occurred at the age of 18 years. It has been reported MPS can developed in children and young adults. Thirumal and colleagues described 24 cases of MPS between the age of 5 to 11 years. All of them presented with rectal bleeding and about 8% had rectal polyps [11]. In another study that reported 21 cases of MPS between the age of 8 to 16 years, rectal bleeding was also observed in all cases. Rectal polyps were identified in about a quarter of cases. Histologically, fibromuscular hyperplasia in the lamina propria was seen in all the cases, while half of them had crypt abnormalities and a quarter had villiform surface [9].

The histological mimics of MPS with polyp can be divided into those with (1) granulation tissue and (2) glands with intraluminal serration. The differential diagnosis of MPS with polyp containing granulation include juvenile polyp and benign inflammatory polyp in inflammatory bowel disease. While the differential diagnosis of MPS with polyp demonstrating glandular intraluminal serration includes hyperplastic polyp and serrated adenoma (Figure 4).

### 4.1. Juvenile Polyp versus MPS

Juvenile polyp (JP) is a hamartomatous malformation that can be sporadic or hereditary (juvenile polyposis syndrome). Similar to MPS, juvenile polyp can be single or multiple, with presence of granulation tissue and prominent dilated crypts. In contrast to MPS, surface erosion in JP is not common. There is no prominent muscularis mucosa. JP may have prominent inflammatory infiltrates and may involve other part of colon [2].

### 4.2. Benign Inflammatory Polyp versus MPS

Benign inflammatory polyp is often related to inflammatory bowel disease. Similar to MPS, it may be single or multiple, with surface erosion or ulceration and granulation tissue, and crypt distortion or dilation. In contrast to MPS, it may have heavy acute and chronic inflammatory infiltrates with cryptitis and crypt abscess. Benign inflammatory polyp can be located anywhere in the colorectum.

### 4.3. Hyperplastic Polyp and Serrated Adenoma versus MPS

Both hyperplastic polyp and serrated adenoma demonstrate glandular intraluminal serrations or saw-tooth appearance, which can also be seen in MPS. However, hyperplastic polyp and serrated adenoma lack other features observed in MPS, like surface ulceration and granulation tissue. In addition, these polyps can be located anywhere in the colorectum. MPS may also be mistaken as adenocarcinoma due to displacement of crypts in the muscularis mucosa and submucosa [2].

## 5. Conclusions

The histological diagnosis of MPS could be challenging due to various mimicry. However, the diagnosis can be made with careful identification of its classic histological features. These 2 cases demonstrated some of the spectrum of the histological changes in MPS with polyps. If diagnosis is difficult, we suggest performing a few deeper sections and an elastin stain to demonstrate elastin fibers in the lamina propria. Sometimes the excision is too superficial and the muscularis mucosa is unable to determine, then a comment of repeat biopsy of deeper tissue could be helpful.

## Figures and Tables

**Figure 1 diagnostics-12-00966-f001:**
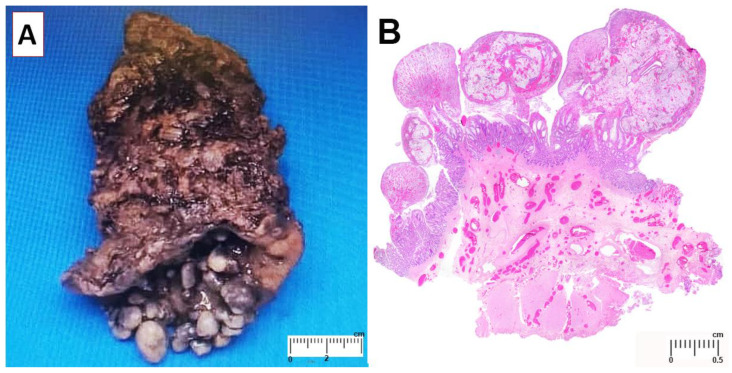
(**A**) The rectal tissue showed multiple polyps covered by mucous. (**B**) Histologically, the polyps consist of granulation tissue. The muscularis mucosa is hypertrophic with splayed fibers (H&E).

**Figure 2 diagnostics-12-00966-f002:**
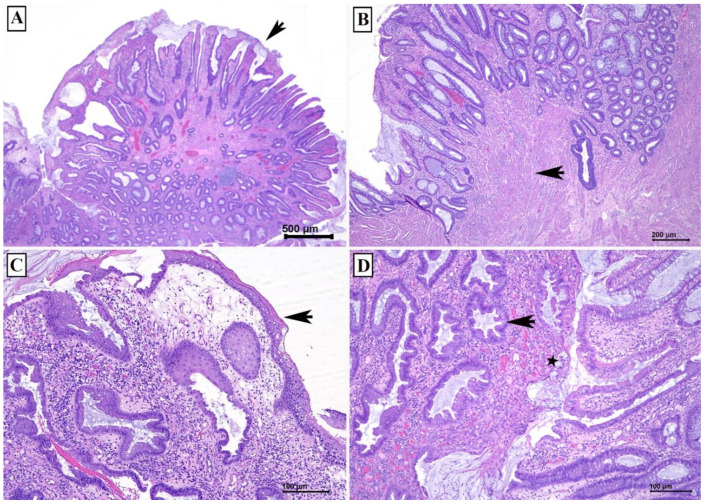
Classic histological features of mucosal prolapse syndrome. (**A**) Polyp demonstrating villiform surface and covered by exudate (arrow) (H&E, scale bar 500 µm). (**B**) Thickened and splayed muscularis mucosa (arrow) (H&E, scale bar 200 µm). (**C**) Focal area covered by squamous epithelium (arrow) (H&E, scale bar 100 µm). (**D**) The glands demonstrate intraluminal serration (arrow) with presence of granulation tissue (star) (H&E, scale bar 100 µm).

**Figure 3 diagnostics-12-00966-f003:**
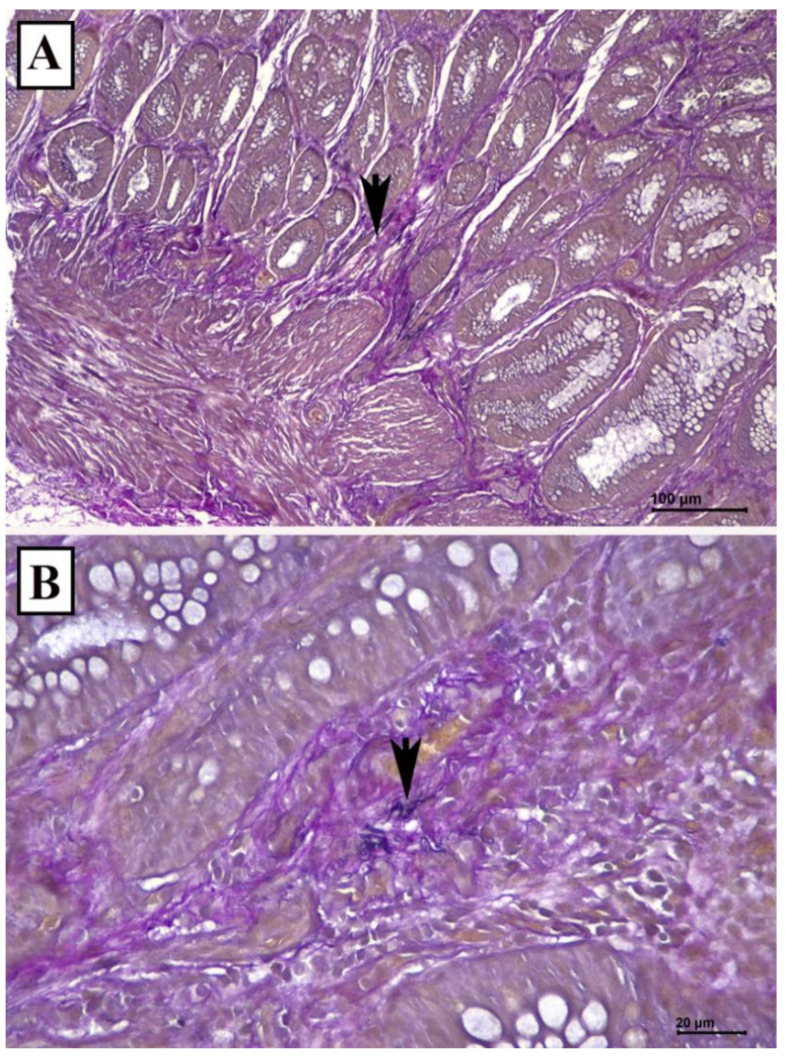
Weigert Van Gieson stain demonstrates elastic fibres in the lamina propria (arrow) (**A**): scale bar 100 µm, (**B**): scale bar 20 µm).

**Figure 4 diagnostics-12-00966-f004:**
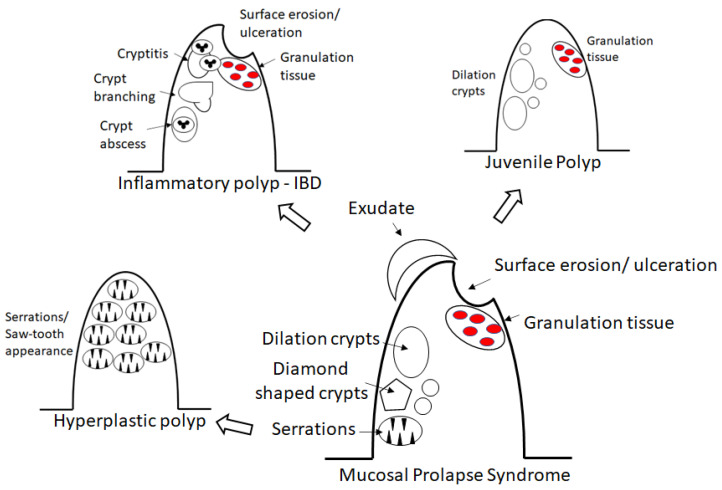
A diagram demonstrating the similarity and differences of mucosal prolapse syndrome with its mimics.

## Data Availability

Not applicable.

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
