# Peer review of "Rectal Polyposis in Mucosal Prolapse Syndrome"

_diagnostics, 2022, doi:10.3390/diagnostics12040966_

Round 1

Reviewer 1 Report

Firstly, to the best of my knowledge, there is a spectrum of mucosal prolapse syndromes, which is group of benign chronic inflammatory diseases that includes rectal prolapse, solitary rectal ulcer syndrome (SRUS), proctitis cystica profunda (PCP), inflammatory cloacogenic polyp, inflammatory cap polyps and inflammatory myoglandular polyps. Therefore, the nomenclature used in the manuscript is not sufficiently clear and in line with published data. According to this, the discussion should be revised, alongside Figure 4. 

Secondly, the information provided on patients is scarce- there is no clear  mention of possible constipation in presented cases (and the aforementioned disorders are to be linked to linked to irregular bowel movements), presence of pain, other laboratory findings (especially albumin- polyposis has been previously linked to hypoalbuminemia, C-reactive protein and other markers of inflammation), as well as microbiological work-up. Histopathological features should be described in more detail (layer by layer) in case presentations. 

Author Response

Response to reviewers’ comments

We would to thank the reviewers for the constructive and valuable comments.

Reviewer 1

Firstly, to the best of my knowledge, there is a spectrum of mucosal prolapse syndromes, which is group of benign chronic inflammatory diseases that includes rectal prolapse, solitary rectal ulcer syndrome (SRUS), proctitis cystica profunda (PCP), inflammatory cloacogenic polyp, inflammatory cap polyps and inflammatory myoglandular polyps. Therefore, the nomenclature used in the manuscript is not sufficiently clear and in line with published data. According to this, the discussion should be revised, alongside Figure 4. 

Response: Thank you for the comments. We agree mucosal prolapse syndrome is a spectrum of diseases that include rectal prolapse, solitary rectal ulcer syndrome (SRUS), proctitis cystica profunda (PCP), inflammatory cloacogenic polyp, inflammatory cap polyps and inflammatory myoglandular polyps. A new sentence was added to describe the spectrum of mucosal prolapse syndrome. See page 3, line 103. Inflammatory cap polyposis has been removed as it should not be considered as mimic as suggested (See manuscript text page 5 and figure 4).

Secondly, the information provided on patients is scarce- there is no clear  mention of possible constipation in presented cases (and the aforementioned disorders are to be linked to linked to irregular bowel movements), presence of pain, other laboratory findings (especially albumin- polyposis has been previously linked to hypoalbuminemia, C-reactive protein and other markers of inflammation), as well as microbiological work-up. Histopathological features should be described in more detail (layer by layer) in case presentations. 

Response: Additional history and laboratory findings were added. Both cases did not have infection with normal white cell count and CPR was performed in one of them which was also normal. Both cases also had normal albumin level. See page 1 and 2. The positive histopathological findings have been discussed and we can unsure how to further expand it.

Reviewer 2 Report

I recommend the publication of manuscript “Rectal Polyposis in Mucosal Prolapse Syndrome” in Diagnostics after minor revision. In my opinion, English language is understandable, and the work does not require any editing. Authors noticed some interesting observations in two cases of mucosal prolapse syndrome presented as rectal polyposis and provide a discussion on its histological differential diagnosis.

Below I present what should be improved at this manuscript:

  1. Please expand the Introduction. This fragment, should introduce the Readers in the topic and theory, certainly should be longer than the Abstract.
  2. Please put the scale bar to the Fig 1A so that the Reader can relate the size of rectal tissue to the scale.
  3. It is a good practice to publish histological photos with a scale bar next to photos. Please, if it’s possible also include the scale on photos Fig1B, Fig.2, Fig.3. If not, please explain why it is impossible.
  4. Please add a detailed description of the staining method - methodology of histological examination, and information on which microscope the photos were taken (company, model).
  5. Did the patients from the described cases sign their consent to participate in the study? or whether the study was based on the consent of the bioethics committee? please add this information in the manuscript.
  6. References include 10 publications, some of them are very out of date, I understand that there is a need for citing specific studies, but where possible, please quote more recent studies (max. 5 years), if not, please explain why it is impossible .

This recommendations only refines the overall draft of the manuscript.

The manuscript submitted by the Authors is in line with the subject of the Diagnostics, and will be an attractive article for the Readers.

Author Response

Response to reviewers’ comments

We would to thank the reviewers for the constructive and valuable comments.

Reviewer 2

I recommend the publication of manuscript “Rectal Polyposis in Mucosal Prolapse Syndrome” in Diagnostics after minor revision. In my opinion, English language is understandable, and the work does not require any editing. Authors noticed some interesting observations in two cases of mucosal prolapse syndrome presented as rectal polyposis and provide a discussion on its histological differential diagnosis.

Below I present what should be improved at this manuscript:

  1. Please expand the Introduction. This fragment, should introduce the Readers in the topic and theory, certainly should be longer than the Abstract.

Response: Thanks for the comment. The introduction is expanded with additional information on pathogenesis and clinical presentation of MPS. A reference (2017) is added.

See introduction, Page 1

The incidence of MPS is approximately 1 in 100,000 with young adult predominant. The proposed pathogenesis of SRUS is due to excessive straining which results in high intrarectal pressure and ischaemia. The anterior rectal mucosa is forced into the closed anal canal due to contracting/ non-relaxation of puborectalis muscle, leading to congestion and oedema, forming ulcer or polyp. It may present clinically with bleeding, mucus discharge, abdominal pain, tenesmus, feelings of incomplete defaecation and constipation [1].

Abreu, M.; Azevedo Alves, R.; Pinto, J.; Campos, M.; Aroso, S. Solitary Rectal Ulcer Syndrome: A Paediatric Case Report. GE Port J Gastroenterol. 2017;24(3):142-146.

  1. Please put the scale bar to the Fig 1A so that the Reader can relate the size of rectal tissue to the scale.

Response: Scale added to Fig 1A.

See figure 1, page 2

  1. It is a good practice to publish histological photos with a scale bar next to photos. Please, if it’s possible also include the scale on photos Fig1B, Fig.2, Fig.3. If not, please explain why it is impossible.

Response: Scale bar added to all the figures in Fig 1B, Fig. 2 and Fig. 3.

See figure 1, 2 and 3, page 2-4.

  1. Please add a detailed description of the staining method - methodology of histological examination, and information on which microscope the photos were taken (company, model).

Response: The elastic fiber staining protocol and product information are added. In addition, the information on the camera used to obtain all the histological figures and the microscope used were also added.

See Page 3. Line 86.

  1. Did the patients from the described cases sign their consent to participate in the study? or whether the study was based on the consent of the bioethics committee? please add this information in the manuscript.

Response: Yes, consent has been obtained from the patients. See page 6, line 178.

  1. References include 10 publications, some of them are very out of date, I understand that there is a need for citing specific studies, but where possible, please quote more recent studies (max. 5 years), if not, please explain why it is impossible.

Response: Thank you for the comments. Reference no 7, 9 and 10 were recently published in 2020 and 2021. Some are very old, for example reference no 3 was from 1829. However, this is to describe the first time it was discovered/described. We have added another article published in 2017 in the introduction.

This recommendations only refines the overall draft of the manuscript.

The manuscript submitted by the Authors is in line with the subject of the Diagnostics, and will be an attractive article for the Readers.

Response: Thank you for the comment.

Round 2

Reviewer 1 Report

No further comments.